# The Role of Distinctive Sphingolipids in the Inflammatory and Apoptotic Effects of Electronegative LDL on Monocytes

**DOI:** 10.3390/biom9080300

**Published:** 2019-07-24

**Authors:** Núria Puig, Montserrat Estruch, Lei Jin, Jose Luis Sanchez-Quesada, Sonia Benitez

**Affiliations:** 1Cardiovascular Biochemistry, Biomedical Research Institute Sant Pau (IIB-Sant Pau), Barcelona, Spain. C/Sant Quinti 77-79, 08041 Barcelona, Spain; 2Department of Biochemistry and Molecular Biology, Faculty of Medicine, Building M, Universitat Autònoma de Barcelona (UAB), Cerdanyola del Vallès, 08193 Barcelona, Spain; 3CIBER of Diabetes and Metabolic Diseases (CIBERDEM), 28029 Madrid, Spain

**Keywords:** electronegative LDL, ceramide, sphingosine, inflammation, apoptosis, monocytes

## Abstract

Electronegative low-density lipoprotein (LDL(−)) is a minor LDL subfraction that is present in blood with inflammatory and apoptotic effects. We aimed to evaluate the role of sphingolipids ceramide (Cer), sphingosine (Sph), and sphingosine-1-phosphate (S1P) in the LDL(−)-induced effect on monocytes. Total LDL was subfractioned into native LDL and LDL(−) by anion-exchange chromatography and their sphingolipid content evaluated by mass spectrometry. LDL subfractions were incubated with monocytes in the presence or absence of enzyme inhibitors: chlorpromazine (CPZ), d-erythro-2-(*N*-myristoyl amino)-1-phenyl-1-propanol (MAPP), and *N*,*N*-dimethylsphingosine (DMS), which inhibit Cer, Sph, and S1P generation, respectively. After incubation, we evaluated cytokine release by enzyme-linked immunosorbent assay (ELISA) and apoptosis by flow cytometry. LDL(−) had an increased content in Cer and Sph compared to LDL(+). LDL(−)-induced cytokine release from cultured monocytes was inhibited by CPZ and MAPP, whereas DMS had no effect. LDL(−) promoted monocyte apoptosis, which was inhibited by CPZ, but increased with the addition of DMS. LDL enriched with Sph increased cytokine release in monocytes, and when enriched with Cer, reproduced both the apoptotic and inflammatory effects of LDL(−). These observations indicate that Cer content contributes to the inflammatory and apoptotic effects of LDL(−) on monocytes, whereas Sph plays a more important role in LDL(−)-induced inflammation, and S1P counteracts apoptosis.

## 1. Introduction

Sphingolipids have an essential structural function in cell membranes. In addition, these lipid metabolites are emerging as important intracellular and intercellular signaling molecules that are involved in the progress of inflammatory diseases and atherosclerosis [1]. These bioactive lipids are present in atherosclerostic plaque, where they play inflammatory and apoptotic roles [2]. Sphingolipid metabolism is complex and highly interconnected, and each step is controlled by specific enzymes. The core component of sphingolipid metabolism is ceramide (Cer), which is generated by the hydrolysis of sphingomyelin (SM) by sphingomyelinase activity (SMase), and can be hydrolyzed by ceramidase (CDase) activity, yielding sphingosine (Sph). Sph can subsequently be phosphorylated by S1P kinase to generate sphingosine-1-phosphate (S1P) [3].

Cer is a widely studied molecule that regulates cell differentiation, inflammation, and apoptosis [4]. Its plasma concentration is elevated in patients who are at high cardiovascular risk [5,6]. The role of Sph in atherosclerosis and inflammation is not clearly established, and its biological effects have been reported to overlap with those promoted by Cer [7]. Sph has been shown to promote inflammatory response [8] and apoptosis [9], although it has also been reported to induce an anti-apoptotic effect [10]. S1P is a pleiotropic bioactive metabolite, and is an important intercellular signaling molecule recognized by specific cell receptors whose subtype determines the effect on cells [11]. S1P is reported to promote cell proliferation, survival, and migration [12,13,14]. Regarding inflammation, some studies have described how it promotes inflammation [15,16,17], whereas others have reported an anti-inflammatory action, mainly when it is bound to its major carrier, high-density lipoprotein (HDL) [18,19,20]. Although LDL may contain low quantities of S1P, it mainly carries Cer and Sph [21]. Some studies have shown that LDL with an increased Cer content induces inflammation [22,23], whereas the effect of an increased content of Sph and S1P in LDL is unknown. 

Electronegative low-density lipoprotein (LDL(−)) is a minor LDL subfraction found in blood circulation that has various atherogenic effects, such as the induction of inflammation in cultured cells. It differs from native LDL (LDL(+)) in several physicochemical properties [24], including the presence of phospholipolytic activities. Indeed, LDL(−) has a fivefold to sixfold increased concentration of the enzyme platelet-activating factor acetylhydrolase (PAF-AH) compared with LDL(+) [25]. In addition, a phospholipase C (PLC)-like activity is believed to be associated with LDL(–), having a high avidity to degrade lysophosphatidylcholine (LPC) by lysoPLC activity and sphingomyelin (SM) by SMase activity [26]. These enzymatic activities seem to be involved in the increased concentrations in LDL(−) of Cer and non-esterified fatty acid (NEFA), which are responsible for LDL(–)-induced cytokine release in monocytes [22,27,28]. However, other compounds, such as those resulting from Cer hydrolysis—Sph and S1P—could also be increased in LDL(−) and be involved, not only in the inflammatory effect of LDL(–), but also in a putative apoptotic effect on monocytes. 

LDL(−) exerts an apoptotic effect in macrophages [29] and in endothelial cells [30], being attributed in the latter to the increased Cer content in LDL(−) [30]. In fact, Cer and NEFA are inductors of apoptosis [31,32], and an increased content of these molecules confers apoptotic properties on LDL [31,33,34]. Moreover, modified LDL can promote apoptosis by increasing the Cer content in cells [31,35].

In the current study, we sought to determine the specific involvement of the sphingolipids Cer, Sph, and S1P in the inflammatory and apoptotic effects of LDL(−) in human monocytes isolated from plasma. For this purpose, we used specific inhibitors of SMase, CDase, and Sph kinase activities, as summarized in the Methods section.

## 2. Material and Methods 

### 2.1. LDL Isolation and Modification

Plasma samples from healthy normolipemic subjects (total cholesterol <5.2 mmol/L, triglyceride <1 mmol/L) were obtained in EDTA-containing Vacutainer tubes. All the subjects gave their written informed consent, and the study was conducted after approval from the Institutional Ethics Committee of the Hospital Sant Pau (IIBSP-APO-2013-105, 25 February 2015). All the LDL preparations were performed in conditions that prevented oxidation and endotoxin contamination [36]. LDL (1.019–1.050 kg/L) was isolated from plasma by sequential flotation ultracentrifugation at 4 °C. In some experiments, LDL was modified by different treatments: by incubation with SMase (10 U/g apoB) for 2 h at 37 °C to increase its Cer concentration (SMase-LDL); by incubation with NEFA (NEFA-LDL) [28]; and by incubation with Sph-enriched liposomes (10 μM) (Sph-LDL), as described for Cer-enriched liposomes [22]. The enrichment in these compounds was assessed by thin-layer chromatography, and the aggregation level of modified LDL was evaluated by gradient gel electrophoresis, as previously described [22].

Total LDL was fractionated in LDL(+) and LDL(−) by preparative anion-exchange chromatography in an ÄKTA-FPLC system (GE Healthcare) [37]. LDL fractions were concentrated with Amicon centrifugal filters (Merck Millipore). LDL(+) and LDL(−) compositions were determined in a Cobas^®^ 501 autoanalyzer, including total cholesterol, triglyceride, apoB (Roche, Basel, Switzerland), NEFA, and phospholipid (Wako, Neuss, Germany).

### 2.2. Lipidomic Analysis

Baseline levels of sphingolipids in LDL(+) and LDL(−) were evaluated by liquid chromatography-mass spectrometry (LC-SM) in the CIBERDEM-Metabolomics Platform of Universitat Rovira i Virgili. First, a lipid extraction of LDL samples (32 µg apoB) was performed. LDLs were lyophilized and resuspended in 220 μL of methanol (MeOH) followed by vortexing. Then, 440 μL of dichloromethane were added followed by vortexing. Afterwards, 140 μL of ultrapure water was added, vortexed, and stored at room temperature for 20 min. The solution was centrifuged for 10 min at 14,500 rpm (4 °C). Next, 400 µL of the organic phase was collected and evaporated to dryness with N_2_. Samples were reconstituted in 150 µL of MeOH:toluene (9:1) and transferred to LC-MS vials. Lipid extracts (5 μL) were injected in a Ultra High Perfomance Liquid Chromatography system (1290 Agilent, Santa Clara, CA, USA) coupled to a triple quadrupole (QqQ) mass spectrometer (6490 Agilent Technologies) operated in positive electrospray ionization (ESI+) mode. The instrument was set to acquire in multiple reaction monitoring mode. Lipids were separated using C18-RP (ACQUITY UPLC BEH 2.1 × 150 mm, 1.7 μm, Waters) chromatography at 65 °C and at a flow rate of 0.4 mL/min. The solvent system was A  =  acetonitrile:water (60:40) in 10 mM of ammonium formate, and B  =  isopropanol:acetonitrile (90:10) in 10 mM of ammonium formate. The gradient elution started at 15% B and went to 30% from minute 0 to 2, 48% B from minute 2 to 2.5, 82% B from minute 2.5 to 11, and 99% B from minute 11 to 11.5. Quality control samples (QC) consisting of pooled samples were injected before the first study sample and then periodically after four study samples. 

Cer 14:0, Cer 16:0, Cer 18:0, Cer 20:0, Cer 22:0, Cer 24:0, Sph 18:1, and S1P 18:1 relative concentrations were analyzed by this method. Total Cer was considered to be the sum of all the Cer species evaluated.

### 2.3. Inflammatory Action on Monocytes

Peripheral blood from volunteers was collected, and mononuclear cells were isolated by density gradient centrifugation using a density solution of 1.077 kg/L (Lympholyte Cedarlane) [37]. Cells were seeded on 12-well plates (2 × 10^6^ cells/mL), and monocytes were separated from lymphocytes according to their adhesive properties. All the subjects gave their written informed consent, and the study was conducted after approval from the Institutional Ethics Committee of the Hospital Sant Pau.

To evaluate the role of sphingolipids in the inflammatory and apoptotic effects of LDL(−), several inhibitors were tested: chlorpromazine (CPZ) (Sigma) is an inhibitor of SMase activity; d-erythro-2-(*N*-myristoyl amino)-1-phenyl-1-propanol (MAPP) (Cayman) is an inhibitor of neutral/alkaline CDase activity; and *N*,*N*-dimethylsphingosine (DMS) (Cayman) is an inhibitor of Sph kinase. The effect of these inhibitors is summarized in Table 1.

All the inhibitors were used at 10 μM. LDL(+) and LDL(−) were incubated in the presence or absence of these inhibitors for 20 h; afterwards, they were dialyzed against RPMI 1640 medium (supplemented with 1% fetal calf serum, 2 mM of L-glutamine, 0.1 U/L of streptomycin, and 0.1 U/L of penicillin), and filtered in sterile conditions. Then, monocytes were incubated for 20 h with these LDL subfractions (60 mg apoB/L) (pre-incubation conditions). Monocytes were also incubated simultaneously with inhibitors plus LDL(+) or LDL(−) (60 mg apoB/L) (co-incubation), and without LDL (blank or basal condition) in the presence or absence of inhibitors. All the incubations were performed in the above-mentioned RPMI 1640 medium. Cell supernatants were collected after 20 h to evaluate the release of IL1β, IL6, IL10, and MCP1, using ELISA kits (eBioscience for all, except Diaclone for IL1β), as described [38], and following the manufacturer’s recommendations. Moreover, monocytes were collected to analyze apoptosis by flow cytometry. 

### 2.4. Apoptosis Assay

Apoptosis assays were performed in monocytes incubated with LDL(−) plus MAPP or DMS, and with LDL(−) pre-incubated with CPZ (all inhibitors at 10 µM). Apoptosis was determined by an Annexin V-FITC, Apoptosis detection kit (eBioscience-Invitrogen, Carlsbad, CA, USA). Briefly, monocytes were collected and resuspended in 200 μL of binding buffer (200,000 cells). Cell suspension was stained with Annexin V-FITC, which binds to phosphatidylserine, for 10 min in darkness at room temperature. Afterwards, cells were washed and resuspended in 190 µL of binding buffer. After the addition of 10 μL of propidium iodide (PI), the fluorescence on stained cells was measured by flow cytometry in a MACSQuant^®^ Analyzer MQ10 (Milteny Biotech GmbH, Bergisch Gladbach, Germany) and analyzed using MACSQuant software (Milteny Biotech GmbH). Non-stained cells (A^-^PI^-^) were considered to be viable cells, A^+^PI^-^ was considered to indicate early apoptosis, double staining (A^+^PI^+^) was considered to indicate late apoptosis, and A^-^PI^+^ was considered to indicate necrotic cells. Total apoptosis is the sum of early and late apoptosis.

### 2.5. Statistical Analyses

Differences between groups were tested using the Wilcoxon signed-rank test for paired data. Results were expressed as mean ± SD. A value of *p* < 0.05 was considered to be statistically significant.

## 3. Results

### 3.1. LDL(−) Has an Increased Content in Sph and Cer

We aimed to evaluate the content of Cer, Sph, and S1P in LDL(+) and LDL(−) by mass spectrometry. Figure 1 shows that LDL(−) had a much higher content in Cer than LDL(+), as previously described by a less sensitive method [22]. In addition, LDL(−) showed a twofold content in Sph compared to LDL(+), whereas S1P was undetectable in both LDL subfractions. 

### 3.2. Sph and Cer Are Involved in the Cytokine Release Promoted by LDL(−) in Monocytes

The role of Cer, Sph, and S1P in the cytokine release promoted by LDL(−) in monocytes was evaluated. For this purpose, specific inhibitors involved in the generation of the above-mentioned sphingolipids were used, respectively: chlorpromazine (CPZ) (SMase inhibitor), d-erythro-2-(*N*-myristoyl amino)-1-phenyl-1-propanol (MAPP) (CDase inhibitor), and *N*,*N*-dimethylsphingosine (DMS) (Sph kinase inhibitor). 

Inhibitors alone did not modify the release of cytokines compared to basal values in the absence of LDLs (data not shown). 

In the presence of LDL(−), the addition of DMS did not alter the release of monocyte chemoattractant protein 1 (MCP1) and interleukin (IL)-6 promoted by LDL(−), thereby suggesting that S1P generation was not involved in the inflammatory action of this lipoprotein (Figure 2). By contrast, the addition of CPZ inhibited the release of these cytokines induced by LDL(−).

Finally, the addition of MAPP decreased MCP1 and IL-6 release induced by LDL(−) as well as that of IL-1β and IL-10, in a concentration-dependent manner (Figure 3). However, the effect promoted by LDL(+) was not modified by the addition of any inhibitor (in the case of MAPP, inhibition lower than 5%). Taken together, the results indicate that both Sph and Cer contribute to LDL(−)-induced inflammation in human monocytes.

To confirm the role of increased Sph and Cer content in LDL on the inflammatory response of monocytes, we conducted experiments in which LDL(+) was enriched “in vitro” with these compounds by incubation with liposomes containing Sph or by SMase lipolysis, respectively. Appendix A shows the enrichment of Sph and Cer in these “in vitro”-modified LDLs and the effect of such modification in the aggregation of LDL particles. The increase in Sph and Cer was 1.5-fold and sixfold, respectively, as calculated by densitometry versus unmodified LDL. Figure 4 shows that compared to unmodified LDL, LDL enriched with Sph (Sph-LDL), and LDL treated with SMase (SMase-LDL) induced the release of IL-6, as a representative cytokine induced by LDL(−).

Next, we sought to determine whether the inhibition induced by MAPP and CPZ on cytokine release was due to a direct effect on LDL(−) or was a consequence of an effect on cell enzymes. For this purpose, LDL(−) was pre-incubated with MAPP or CPZ, and then dyalized to remove excess inhibitor prior to its addition to monocytes. This effect was compared to that promoted when monocytes were incubated simultaneously with LDL(−) and an inhibitor (co-incubation). Figure 5 shows IL-6 release, as a representative cytokine induced by LDL(−). No significant difference was observed when using MAPP between pre-incubation and co-incubation. By contrast, CPZ exerted a more effective inhibitory effect in co-incubation than in pre-incubation conditions. These observations indicate that CPZ has a direct effect on cells, perhaps by inhibiting cell SMase.

### 3.3. Cer Is Involved in LDL(−)-Induced Apoptosis

Figure 6 shows that LDL(−) displayed a twofold increase in the total apoptotic effect on monocytes compared to LDL(+), which did not promote apoptosis. LDL(−) did not induce statistically significant differences in necrosis compared to LDL and the blank of cells.

The effect of enzyme inhibitors was also assayed in LDL(−)-induced apoptosis. Figure 7 shows that DMS enhanced the apoptotic effect of LDL(−), whereas CPZ decreased this effect, both in early and late apoptosis. These observations suggest that Cer is involved in LDL(−)-induced apoptosis, whereas S1P would tend to counteract this action. On the other hand, the presence of MAPP did not promote any significant effect, despite a slight tendency to increase late apoptosis. 

However, MAPP and CPZ alone did not promote apoptosis in the absence of LDL(−), whereas DMS did (Appendix A). 

LDL(+) was modified to increase its content in Cer, Sph, and NEFA, in order to reproduce the apoptotic behavior of LDL(−). Figure 8 shows that although NEFA-LDL and Sph-LDL promoted a slight non-significant apoptotic effect, the highest and statistically significant effect on total apoptosis was found when Cer content in LDL increased after SMase treatment. 

## 4. Discussion

Sphingolipids and their metabolizing enzymes are described to be involved in atherogenesis. This study provides a novel contribution to understanding of the role of sphingolipids in inflammation and apoptosis, particularly focusing on the involvement of these metabolites in the atherogenic effects promoted by LDL(−) on monocytes. As determined by lipidomic analysis, LDL(−) had a higher content in Cer and Sph than LDL(+), and our findings show that the increase in these sphingolipids contributes to its inflammatory and apoptotic effects. Cer content in LDL(−) contributes to both LDL(–)-induced inflammatory and apoptotic effects, whereas Sph plays a more important role in its inflammatory effect. By contrast, S1P counteracts the apoptotic action exerted by LDL(–). These findings are summarized in Figure 9, which shows the effect of the inhibitors used in the study on sphingolipid generation and the ensuing LDL(–)-induced inflammatory and apoptotic actions. 

The increased Cer content in LDL(−) was previously determined by thin layer chromatography, which is a much less sensitive method [22]. The current study corroborated this observation and also showed a higher content of Sph in LDL(−) compared to LDL(+). On the other hand, we found undetectable basal levels of S1P in both LDL fractions. Other authors have detected very low levels of S1P in LDL [21,39]. According to Hammad et al., the distribution of S1P in plasma lipoproteins was of 94.94% in HDL and only 3.73% in LDL. Compared to that study, the absence of S1P in our LDL subfractions could be due to different experimental conditions, such as the range of isolation of LDL and the lower concentration of the sample.

The inflammatory action of LDL(−) on monocytes was reported to be promoted, in part, by its increased content in NEFA and Cer [22,28]. The increased Cer content was proposed to be generated by the SMase activity ascribed to LDL(−) [22]. This hypothesis is supported by the current data, with the use of the SMase inhibitor CPZ, which inhibits the cytokine release promoted by LDL(−) in monocytes. Since CPZ blocks the degradation of sphingomyelin to yield Cer, it probably also blocks the generation of sphingolipids derived from Cer hydrolysis. In fact, CPZ was the inhibitor that promoted the greatest decrease in inflammation and apoptosis. The effect of CPZ was greater in co-incubation experiments with monocytes and LDL(−) added simultaneously, thereby suggesting that CPZ could block not only the intrinsic SMase activity in LDL(−), but also cell enzymatic activity. 

The role of Sph in inflammation is largely unknown, although it has been reported that it induces inflammasome activation, leading to IL-1β induction in macrophages [8]. The current study shows that Sph contributes to LDL(−)-induced inflammation in monocytes. This is suggested by the inhibition exerted by MAPP on the cytokine release promoted by LDL(−). However, MAPP is a CDase inhibitor, which hinders Sph and NEFA generation; thus, its inhibitory effect could also be partly promoted by decreasing NEFA. The inflammatory role of Sph was also suggested by the effect promoted by Sph-LDL. LDL(+), modified to increase its Sph content 1.5-fold, promoted cytokine release to a similar extent to that promoted by LDL(−). Notably, this artificial enrichment was similar to the increased content of Sph found in LDL(−). The higher Sph content in LDL(−) seems to be generated by a CDase-like activity, the origin of which could be cells or LDL(−) itself. The latter is suggested by the lack of difference on LDL(−)-induced cytokine release between pre-incubation and co-incubation with MAPP. 

S1P can display both pro-inflammatory and anti-inflammatory effects, depending on the S1P receptor subtype expressed on cells [40]. However, LDL(−) did not exhibit an increased content in S1P, and DMS did not inhibit LDL(−)-induced cytokine release in monocytes. Therefore, our results reject the involvement of S1P in LDL(−)-induced inflammation.

LDL(−) exerts an apoptotic effect in endothelial cells [41,42] and macrophages [29], but its effect on monocytes had not yet been determined. Some observations made it feasible, because in this cell type, LDL(−) induces molecules related to apoptosis, such as caspase-1 [36], Fas [43], and p38 mitogen-activated protein kinase [38]. Regarding other modified LDLs, oxidized LDL exerts a pro-apoptotic effect in macrophages [31,44]. In monocytes, oxidized LDL can promote both pro-apoptotic and anti-apoptotic effects [45,46], and enzimatically modified LDL promotes apoptosis [47]. In the current study, LDL(−) exerted a slight apoptotic action on monocytes, which is rather lower than its effect found previously on endothelial cells. This could be because the studies of endothelial cells were performed with L5, which only includes the most negative subfraction of LDL(−), and because the highest number of apoptotic cells (more than 40%) was only found when L5 was isolated from dyslipidemic patients [41,48,49]. By contrast, when it was isolated from control subjects, apoptosis was lower than 10% [41], which is a similar value to that found in the current study. In addition, in endothelial cells, the effect was mediated through lectin-like oxidized low-density lipoprotein receptor-1 which is a receptor that is barely expressed in monocytes. The induction of apoptosis could lead to different consequences, depending on the cell type, which is considered to have a pro-atherogenic effect in endothelial cells, whereas in monocytes, it could be hypothesized that it is a physiological control mechanism to decrease the accumulation of inflammatory cells. 

Some molecules whose concentration is increased in LDL(−), such as LPC, Cer, and NEFA [22,50], have been shown to be involved in apoptosis [2,32,51]. The results of the current study strongly suggest that Cer content is mainly responsible for the apoptosis promoted by LDL(−) on monocytes, which is in agreement with that reported for endothelial cells [30]. First, the inhibition of SMase activity decreased LDL(−)-induced apoptotic activity on monocytes to an extent similar to that promoted by LDL(+); second, an increased Cer concentration in LDL(+), which was promoted by SMase treatment, induced apoptosis on monocytes at the same level as that promoted by LDL(−).

It has been reported that Sph is able to exert apoptotic effects [9,10]. In our experiments, Sph was not involved in the apoptotic effect promoted by LDL(−). This fact could be explained by the overlapping biological effects promoted by MAPP, altering the balance between pro-apoptotic and anti-apoptotic molecules owing to CDase inhibition, because MAPP may exert both an anti-apoptotic effect by decreasing Sph and NEFA content, and an apoptotic effect by increasing Cer content. However, the inhibition of S1P formation by the addition of DMS to monocytes increased LDL(−)-induced apoptosis. This observation is in agreement with the assertion that S1P has anti-apoptotic effects [14].

In summary, our data showed that the increased Cer content in LDL(−) contributed to its inflammatory and apoptotic effects on monocytes. Sph played a key role in inflammation, and S1P counteracted the apoptotic effect of LDL(−). The main limitation of our study is that it is an “in vitro” approximation, and it is difficult to evaluate the effect of the different sphingolipids separately, owing to the complexity of their metabolism and their interconversion. In addition, other factors remain to be examined in future studies, including the presence of enzymatic activities in LDL(−), yielding its increased content in Sph, and the content of this bioactive sphingolipid through incubation with monocytes in the presence or absence of enzyme inhibitors.

## 5. Conclusions

The current study provides new knowledge regarding the role of sphingolipids in mediating LDL(−)-induced inflammatory and apoptotic effects in monocytes. Our data suggest that the increased Cer content in LDL(−) contributes to both effects. Not only Cer, but also its hydrolysis product Sph, seems to play a key role in inflammation, and S1P counteracts the apoptotic effect of LDL(−).

## Figures and Tables

**Figure 1 biomolecules-09-00300-f001:**
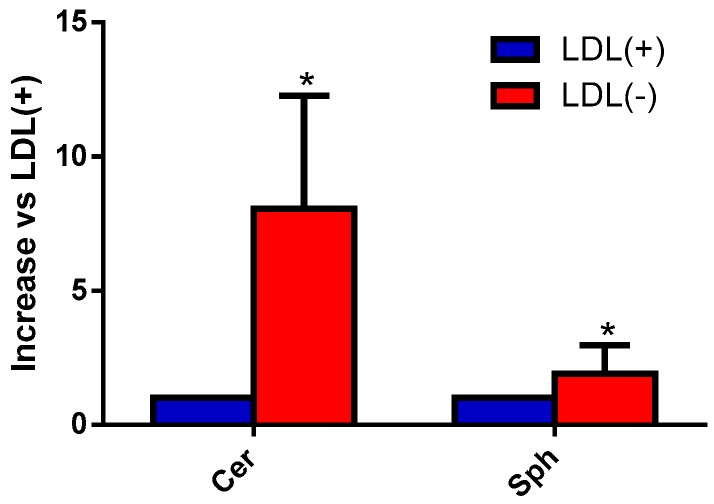
Sphingosine (Sph) and ceramide (Cer) basal content in low-density lipoprotein (LDL) subfractions. Native LDL(+) (blue bars) and negative LDL(−) (red bars) (32 μg of apolipoprotein B (apoB) were analyzed by mass spectrometry to evaluate the baseline levels of sphingolipids. Cer includes all the Cer species that were evaluated. Results are expressed as a relative increase versus LDL(+), mean ± SD (*n* = 4), * vs. LDL(+), *p* < 0.05.

**Figure 2 biomolecules-09-00300-f002:**
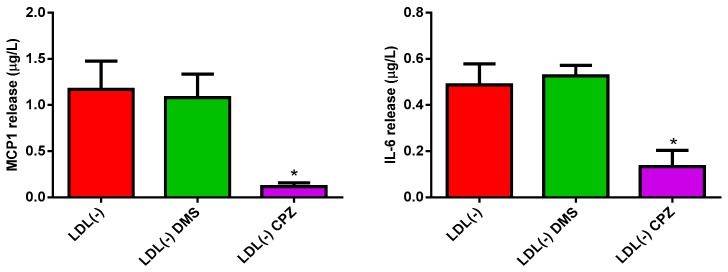
MCP1 and interleukin-6 (IL-6) release promoted by LDL(−) in human monocytes in the presence or absence of DMS and chlorpromazine CPZ. Monocytes were incubated with LDL(−) (60 mg apoB/L) in the absence (red bars) or presence of DMS (green bars) and CPZ (purple bars), both at 10 µM. After 20 hours of incubation, cytokine release was measured by ELISA. Results are expressed as mean ± SD (*n* = 5), * vs. absence of inhibitor, *p* < 0.05.

**Figure 3 biomolecules-09-00300-f003:**
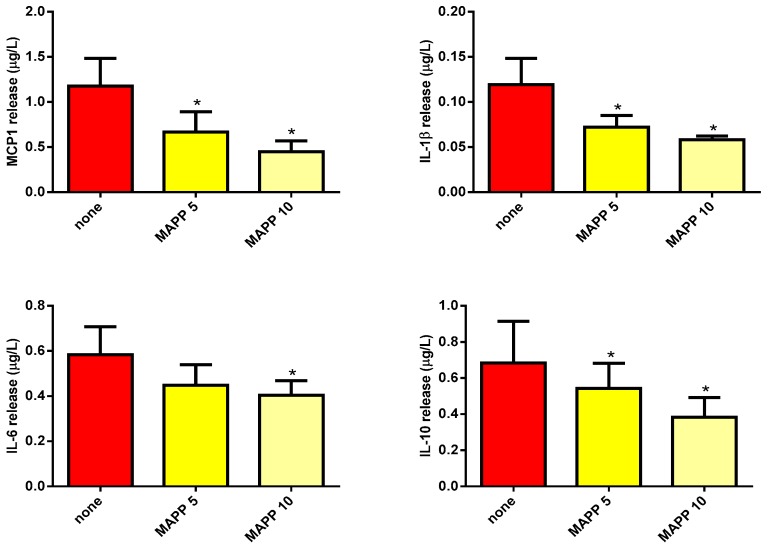
MCP1, IL-1β, IL-6, and IL-10 release promoted by LDL(−) in human monocytes in the presence or absence of MAPP. Monocytes were incubated with LDL(−) (60 mg apoB/L) in the absence of MAPP (red bars) or in the presence of MAPP at 5 µM (yellow bars) and 10 µM (light yellow bars). After 20 hours of incubation, cytokine release was measured by ELISA. Results are expressed as mean ± SD (*n* = 5), * vs. the absence of inhibitor, *p* < 0.05.

**Figure 4 biomolecules-09-00300-f004:**
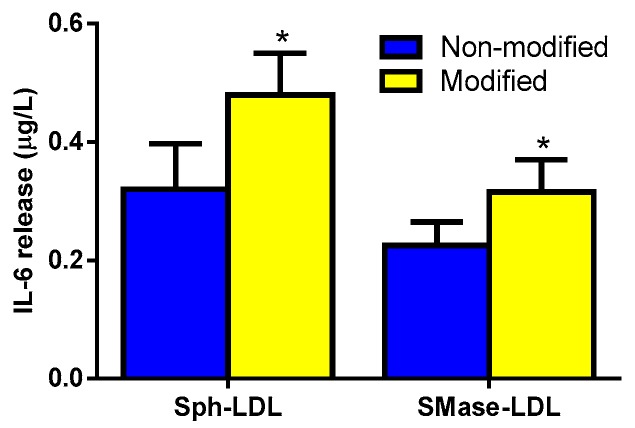
IL-6 release promoted by “in vitro” modified LDL. Monocytes were incubated with unmodified LDL (blue bars) or modified LDL (yellow bars), both at 60 mg apoB/L. LDL was modified by incubation with Sph-enriched liposomes (at 0 and 10 µM) and by sphingomyelinase (SMase) treatment (at 0 and 10 U enzyme/g apoB). After 20 hours of incubation, cytokine release was measured by ELISA. Results are expressed as μg/L and as mean ± SD (*n* = 6), * vs. unmodified LDL, *p* < 0.05.

**Figure 5 biomolecules-09-00300-f005:**
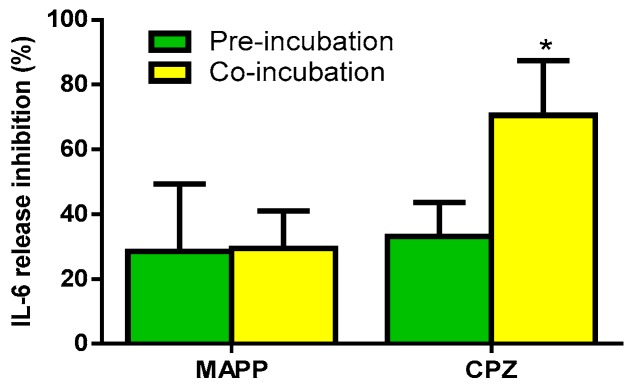
IL-6 release promoted by LDL(−) in pre-incubation or co-incubation with MAPP and CPZ. Monocytes were incubated with LDL(−) (60 mg apoB/L), which was pre-incubated (green bars) or co-incubated (yellow bars) with the inhibitors MAPP and CPZ, both at 10 µM. After 20 h of incubation, cytokine release was measured by ELISA. Results are expressed as % of inhibition versus LDL(−) in the absence of an inhibitor, mean ± SD (*n* = 5), * vs. pre-incubation conditions, *p* < 0.05.

**Figure 6 biomolecules-09-00300-f006:**
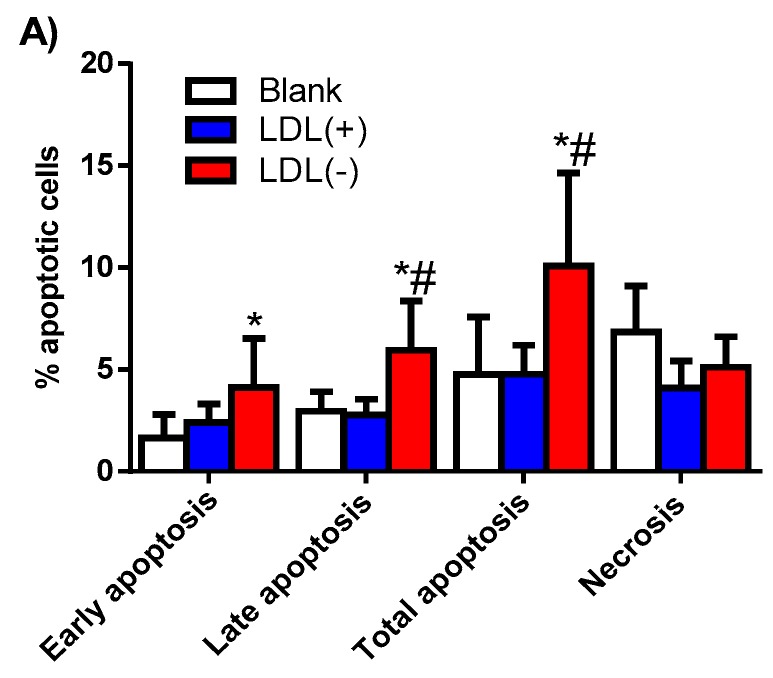
Apoptosis induced by LDL(+) and LDL(−) in human monocytes. (**A**) Early, late, and total apoptosis, and necrosis induced in basal conditions (blank) or by LDL(+) (blue bars) and LDL(−) (red bars) (60 mg apoB/L) were evaluated by flow cytometry. Results are expressed as percentage of apoptotic cells, mean ± SD (*n* = 6), * vs. blank cells, and # vs. LDL(+), *p* < 0.05 (**B**). Representative experiment showing the effect of LDL(+) (left panel) and LDL(−) (right panel) on human monocytes.

**Figure 7 biomolecules-09-00300-f007:**
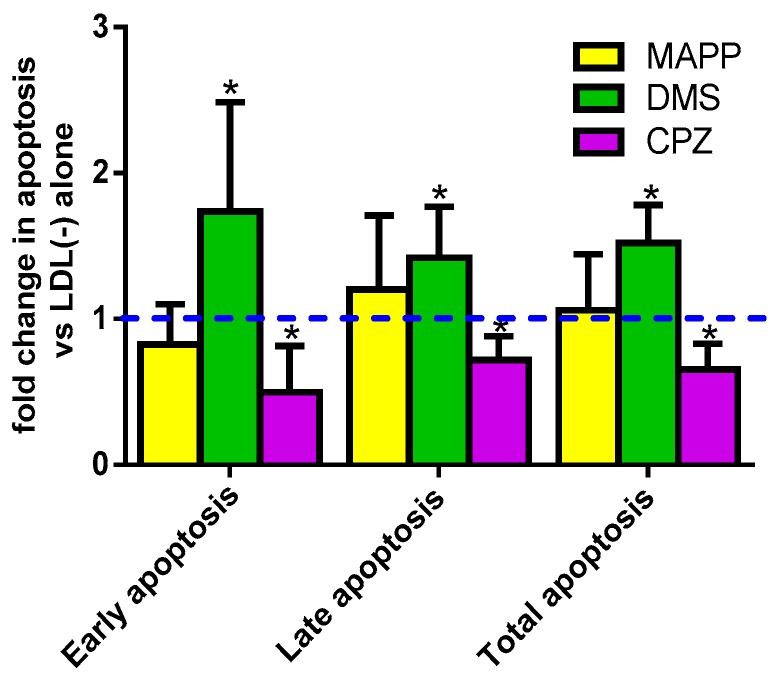
Apoptosis induced by LDL(−) in the presence of inhibitors. Early, late, and total apoptosis induced by LDL(−) (60 mg apoB/L) were evaluated in the absence or presence of MAPP (yellow bars), DMS (green bars), or CPZ (purple bars), all at 10 µM. Results are expressed as fold change in apoptosis versus LDL(−) in the absence of inhibitors, mean ± SD (*n* = 6), * vs. non-inhibitor, *p* < 0.05.

**Figure 8 biomolecules-09-00300-f008:**
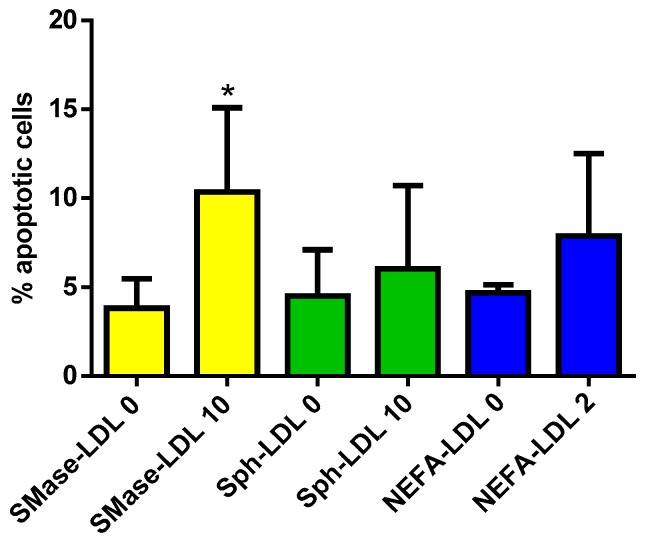
Total apoptosis induced by modified LDL(–). Total apoptosis induced by LDL modified by SMase treatment (yellow bars), Sph (green bars), or non-esterified fatty acid (NEFA) enrichment (blue bars), as described in the Methods section. Results are expressed as mean ± SD (*n* = 5), * vs. unmodified LDL, *p* < 0.05.

**Figure 9 biomolecules-09-00300-f009:**
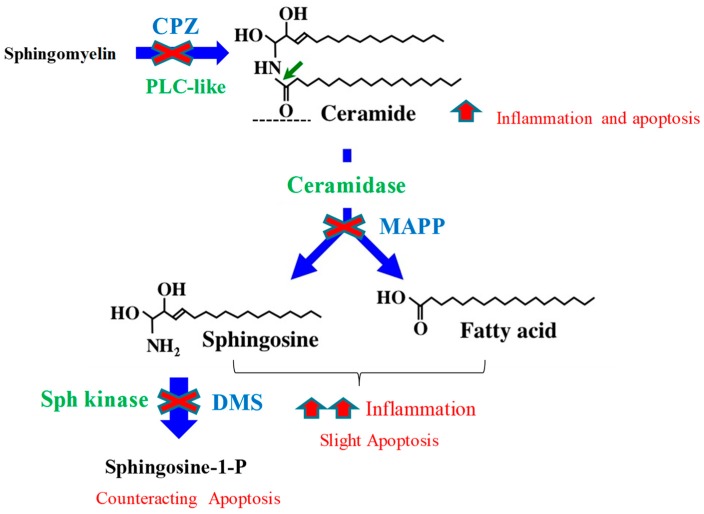
Effect of the inhibitors used in the study on sphingolipid generation and on LDL(−)-induced inflammatory and apoptotic actions. CPZ blocks sphingomyelin degradation, thereby decreasing Cer content and LDL(−)-induced cytokine release and apoptosis. MAPP blocks Cer degradation, leading to a decrease in Sph and NEFA generation, which seems to develop an essential role in inflammation, but a likely minor role in apoptosis. DMS blocks sphingosine-1-phosphate (S1P) generation, thus leading to an increase in the apoptotic effect of LDL(−).

**Table 1 biomolecules-09-00300-t001:** Inhibitors used in the study.

Inhibitor	Abbreviation	Enzyme	Substrate	Product
Chlorpromazine	CPZ	SMase	SM	Cer and phosphorylcholine
d-erythro-2-(*N*-myristoyl amino)-1-phenyl-1-propanol	MAPP	CDase	Cer	Sph and NEFA
*N,N*-dimethylsphingosine	DMS	Sph kinase	Sph	S1P

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
