# Peer review of "The Role of Distinctive Sphingolipids in the Inflammatory and Apoptotic Effects of Electronegative LDL on Monocytes"

_biomolecules, 2019, doi:10.3390/biom9080300_

Round 1
Reviewer 1 Report
In this manuscript, the authors investigated the role of sphingolipids in the inflammatory and apoptotic effects of electronegative LDL on monocytes, and reported that ceramide contributed to both inflammatory and apoptotic effects, sphingosine played a more important role in inflammation, and S1P counteracted apoptosis based on experiments using enzyme inhibitors. Although the manuscript is well written, I think it needs major revision to be published in “Biomolecules”.
[specific comments]
#1. In Figure 1, present the data for the baseline levels of Cer and Sph in LDL(+) and LDL(-) (without incubation for 20 h).
#2. Present the data for the levels of Cer and Sph after incubation for 20 h in the presence or absence of each inhibitor (CPZ, MAPP, DMS).
#3. Since LDH release was not measured in this study, delete its method (page 9, lines 299-300).
#4. What is Wilcoxon T-test (page 10, line 314)?
Reviewer 2 Report
Authors aimed to evaluate role of sphingolipids, in the inflammatory and apoptotic effects of -LDL on monocytes. The manuscript is well written and results are clearly presented.
Major comments: Authors have mentioned that SIP was undetectable in LDL fractions but some studies have shown that SIP can be detectable in LDL fractions! is there any explanation or rationale with the observations? Please update the results with the SIP fractions.
Authors can detect the SIP in plasma and results can be compared
Line 220-221: Unclear. Please consider re writing
Round 2
Reviewer 1 Report
I think that the revised version is much improved compared to the original version and is worth publishing in "Biomolecules".
I will look forward to the following study.